# ML-Dev-Bench: Comparative Analysis of AI Agents on ML development workflows

## Abstract

In this report, we present ML-Dev-Bench, a benchmark aimed at testing agentic capabilities on applied Machine Learning development tasks. While existing benchmarks focus on isolated coding tasks or Kaggle-style competitions, ML-Dev-Bench tests agents' ability to handle the full complexity of ML development workflows. The benchmark assesses performance across critical aspects including dataset handling, model training, improving existing models, debugging, and API integration with popular ML tools. We evaluate three agents - **ReAct**, **Openhands**, and **AIDE** - on a diverse set of 30 tasks, providing insights into their strengths and limitations in handling practical ML development challenges. We open source the benchmark for the benefit of the community.

## 1 Introduction

Recent advances in Large Language Models (LLMs) have demonstrated impressive capabilities in code generation and software engineering tasks. This has led to the development of various benchmarks like HumanEval Chan et al. (2024), MBPP Austin et al. (2021) that evaluate coding abilities, and others like SWE-Bench Jimenez et al. (2024), that test LLM-based agents on software engineering tasks. However, while these benchmarks effectively assess general programming capabilities, they don't capture the unique challenges of Machine Learning development workflows,

Benchmarks like ML-Bench Tang et al. (2024), test agents' abilities to generate code and commands to interact with popular ML repositories, while MLE-Bench Chan et al. (2024) and MLAgentBench Huang et al. (2024) focus on Kaggle-style tasks to evaluate the iterative and open-ended nature of ML development. However, real-world ML development extends far beyond that, including the complexity of working on top of existing codebases and models, integrating with third-party tools, debugging complex issues that span multiple components of the ML pipeline and understanding and balancing trade-offs like model performance and cost to come up with optimal design.

ML-Dev-Bench addresses this gap by providing a comprehensive evaluation framework that tests an agent's ability to handle real-world ML development scenarios. Our benchmark is particularly relevant as ML development increasingly relies on large language models and AI agents to assist developers. Understanding the capabilities and limitations of these agents in handling practical ML development tasks is crucial for their effective deployment in production environments.

## 2 Benchmark Design

ML-Dev-Bench comprises 30 carefully designed tasks that evaluate various aspects of ML development. These tasks are structured to assess both specific technical capabilities (like handling datasets, model implementation) and broader problem-solving skills (like model training and performance improvement) that are essential in real-world ML development. The tasks span several key categories of ML development shown in Table 1:

1. Dataset Handling focuses on evaluating the ability to work with large datasets, inspect them and apply pre-processing pipelines. An example is the noisy imagenette Howard & Others (2019) dataset download task, where the agent needs to download the dataset, inspect its contents to identify the labels file, only load the 50% noisy labels from it and generate class summary statistics.

| Category | Description |
|---|---|
| Dataset Handling | Downloading and preprocessing datasets |
| Model Training | Loading pretrained models, fine-tuning |
| Debugging | Addressing errors in training files, exploding gradients, and incorrect implementations |
| Model Implementation | Modifying and implementing on top of existing model architectures |
| API Integration | Integrating logging tools like WandB |
| Performance | Improving baselines and achieving competitive results |

Table 1: Task Categories and Their Descriptions

| Category | ReAct-Sonnet | OH-Sonnet | Aide-4o | ReAct-4o |
|---|---|---|---|---|
| Dataset Handling | 100% (3/3) | 100% (3/3) | 33% (1/3) | 0% (0/3) |
| Model Training | 67% (4/6) | 83% (5/6) | 33% (2/6) | 50% (3/6) |
| Debugging | 57% (4/7) | 57% (4/7) | 29% (2/7) | 14% (1/7) |
| API Integration | 100% (1/1) | 100% (1/1) | 0% (0/1) | 100% (1/1) |
| Model Implementation | 29% (2/7) | 29% (2/7) | 0% (0/7) | 0% (0/7) |
| Performance | 0% (0/6) | 0% (0/6) | 0% (0/6) | 0% (0/6) |
| **Overall** | **47% (14/30)** | **50% (15/30)** | **17% (5/30)** | **17% (5/30)** |

Table 2: Category-wise Success Rates Across AI Agents

2. Model Training tests an agent's ability to work with existing models, from loading pretrained weights to implementing training loops, logging metrics and managing the training process. These tasks assess both technical skills and the ability to handle long-running tasks.

3. Debugging presents common scenarios including shape errors, exploding gradients, incorrect implementations, and integration errors. Agents must analyze large training logs, metrics, and code across multiple files to identify and resolve issues.

4. Model Implementation tests the ability to modify existing architectures and implement new features. An example is the ChannelViT related tasks, which follow three levels of increasing difficulty: Level 1 provides complete specifications with examples and tests; Level 2 includes specifications and tests but omits examples; Level 3 gives specifications but tests and examples are hidden

5. API Integration assesses the ability to work with essential ML development tools, particularly for logging and experiment tracking.

6. Performance optimization challenges agents to improve baseline implementations through iterative experimentation and hypothesis testing.

## 2.1 EVALUATION METRICS

Tasks are evaluated based on binary success ($\checkmark$) or failure ($\times$). The aggregate success rate for each agent is calculated as Success Rate $= \frac{\text{Total Successful Tasks}}{\text{Total Tasks}} \times 100\%$

Agents are assessed on their ability to complete tasks accurately without introducing errors.

## 3 EVALUATION FRAMEWORK

In this section we briefly describe the design of our evaluation framework, called Calipers, for running the benchmark. The framework consists of three components: agents, evaluation tasks, and

metrics. Agents are evaluated on various Machine Learning tasks to generate metrics. We designed Calipers to allow easy addition of new evaluation tasks, agents and metrics, ensuring the benchmark can evolve alongside advances in ML development practices and tooling.

## 3.1 EVALUATION TASK

Each evaluation task consists of a task description, a set of input code and data files, and a validation logic which checks agent generated outputs and artifacts for correctness. Depending on the type of task, we implement various types of validation checks including

- Running tests on generated code to check for correctness
- Checking for the presence of all required output files and artifacts
- Evaluating agent generated model checkpoints for required performance

## 3.2 AGENTS

Each agent is provided with two inputs in an evaluation run, the description of the task and a working directory populated with initial input files. The agent's outputs are task-specific artifacts which are saved in the working directory. These outputs are validated to determine success or failure. We generate the evaluation metrics discussed in the previous section for each evaluation run. We use litellm callbacks to capture metrics like number of steps, tokens, and cost.

## 4 AGENT SETUP

We evaluate three agents on ML-Dev-bench. The agents and their setup is described below. Each agent uses an LLM and a set of tools to execute various actions. All agents execute their code in a runtime environment which is either a local python or docker environment depending on the agent. We customized the runtime environments for all agents to pre-install common ML frameworks like scikit-learn, pytorch, transformers, lightning, wandb, etc to ensure smooth execution.

1. **ReAct:** We created a simple ReAct agent Yao et al. (2023) as a baseline which takes actions by calling tools. We used the LangGraph framework for the agent and Composio toolset which provides tools for common use cases. We customized the tools to reliably capture large command outputs, handle long running commands and ensure consistency across different tools like file and shell tools. All the tool calls were executed in a local python environment which was pre-installed with common ML frameworks as mentioned earlier and had access to the relevant api keys. No custom prompts were used, and the agent was allowed to run for a maximum of 50 steps. We tested the agent with Claude Sonnet 3.5 10-2022 and OpenAI GPT-4o.
   (a) **Command line tools**
       i. **Shell Tool** - to execute short running commands
       ii. **Spawn Tool** - to execute long running commands like training in the background
       iii. **Sleep and execute tool** - to wait and monitor long running processes
   (b) **File tools** like create files, list files and edit files

2. **Openhands:** Openhands Wang et al. (2024) is a popular open-source coding agent with state-of-the-art performance on SWE-Bench-Full Jimenez et al. (2024). We used Openhands agent v0.21.1 and customized the runtime build to install common ML frameworks listed above. We tested the agent with Claude Sonnet 3.5 10-2022 model which is the current best performing model with the agent on SWE Bench. The agent was allowed to run for a maximum of 50 steps.

3. **AIDE:** AIDE is an agent purpose-built for data science tasks like Kaggle competitions Chan et al. (2024) and performs a tree search over solutions. AIDE scaffolding performs better in comparison to other agents like Openhands on MLEBench using o1, GPT-4o. Unlike other general purpose agents which output any artifact, AIDE outputs an evaluation metric and code as its final output. All other artifacts are considered intermediate outputs and saved in a custom working directory.

# 5 PERFORMANCE COMPARISON

Performance of the agents across different task categories, Table 2 and individual tasks Table 3 reveals a consistent pattern. Performance decreases as tasks become more open-ended and complex. The success rates are highest in well-defined categories like dataset handling and basic debugging, but drop significantly in performance optimization tasks where no agent succeeded. OpenHands-Sonnet (OH-Sonnet) and ReAct-Sonnet are the two best performing agents with 50% and 47% success rate respectively, while AIDE-4o and React-4o achieve 17% success rate.

## 5.1 REACT-SONNET

ReAct-Sonnet achieved a success rate of 47%, demonstrating strong performance in specific, well-defined tasks but struggling with more complex scenarios.

The agent's token usage and costs varied significantly across tasks. Simple operations like dataset downloads cost around $0.02 - 0.08$\$$, while debugging tasks cost between $0.1 - 0.4$\$$, some tasks like ChannelViT-Easy debugging take more steps indicating potentially inefficient exploration in complex scenarios.

A notable strength was ReAct-Sonnet's systematic approach to debugging when provided with specific instructions and test cases. However, the agent showed several limitations such as excessive verification seeking and premature task termination.

## 5.2 REACT-4O

ReAct-4o had some success with tasks with well-defined specifications (WandB logging, downloading a specific model from Torchvision), and certain debugging tasks. However it did struggle on other tasks in the same categories. It also failed on the relatively easier tasks like dataset download due to not following instructions, ran into indentation errors while attempting to debug code and failing to produce output artifacts as required by certain tasks.

## 5.3 OPENHANDS-SONNET

OpenHands-Sonnet demonstrated the highest success rate at 50% (15/30 tasks), showing robust performance across most categories. The agent successfully completed all dataset handling tasks and showed strong performance in model training and debugging.

The agent particularly excelled in structured tasks and showed better persistence in long-running operations. However, it struggled with performance optimization tasks, indicating limitations in open-ended problem-solving scenarios requiring iterative improvement.

## 5.4 AIDE-4O

Aide-4o had a 17% success rate (5/30 tasks), demonstrating limitations across most categories. The agent managed to complete some basic dataset handling and debugging tasks but struggled with model training and completely failed in model implementation and performance optimization categories.

# 6 CONCLUSION

We presented ML-Dev-Bench, a benchmark focused on ML development workflows consisting of 30 tasks. We evaluated 3 agents on this benchmark - ReAct (with Claude Sonnet and GPT-4o), Openhands and AIDE; Openhands with Claude Sonnet performed the best out of these. Future work can involve analysing the impact of scaling compute on these agents; computing variance in success metrics across multiple runs; including reasoning models such DeepSeek-R1, O-1/O-3; and expanding the problem categories to include areas such as label collection. We open-source the evaluation framework for the benefit of the broader community.

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

## A APPENDIX

| Task | Category | ReAct-Sonnet | OH-Sonnet | Aide-4o | ReAct-4o |
|---|---|---|---|---|---|
| Dataset download - Noisy Imagenette | Dataset Handling | ✓ | ✓ | × | × |
| Dataset download - dataset does not exist | Dataset Handling | ✓ | ✓ | ✓ | × |
| Dataset preprocessing | Dataset Handling | ✓ | ✓ | × | × |
| Pretrained model download - Torchvision | Model Training | ✓ | ✓ | ✓ | ✓ |
| Pretrained model download - HuggingFace | Model Training | ✓ | ✓ | × | × |
| Vision finetuning - classification | Model Training | ✓ | ✓ | × | × |
| Overfit on small dataset | Model Training | ✓ | ✓ | ✓ | ✓ |
| Large training logs | Model Training | × | ✓ | × | ✓ |
| CIFAR10 Training | Model Training | × | × | × | × |
| Fix problems in model and dataloader | Debugging | × | × | × | × |
| Model forward pass - shape mismatches | Debugging | ✓ | ✓ | ✓ | ✓ |
| Model Training - shape mismatches | Debugging | ✓ | ✓ | ✓ | × |
| NaN losses | Debugging | ✓ | ✓ | × | × |
| Correct norm for pretrained model | Debugging | ✓ | ✓ | × | × |
| TinyBERT Eval | Debugging | × | × | × | × |
| ViT debugging | Debugging | × | × | × | × |
| Wandb integration | API Integration | ✓ | ✓ | × | ✓ |
| ChannelViT - Easy | Model Implementation | ✓ | ✓ | × | × |
| ChannelViT | Model Implementation | ✓ | ✓ | × | × |
| ChannelViT - No tests | Model Implementation | × | × | × | × |
| VAR implementation | Model Implementation | × | × | × | × |
| Multi-head Latent Attention | Model Implementation | × | × | × | × |
| Multi-head Latent Attention - hidden tests | Model Implementation | × | × | × | × |
| Proximal Policy Optimization | Model Implementation | × | × | × | × |
| Improve CIFAR-10 baseline - existing model ckpt | Performance | × | × | × | × |
| Noisy Imagenette | Performance | × | × | × | × |
| CIFAR-10 long tailed | Performance | × | × | × | × |
| Segmentation | Performance | × | × | × | × |
| BoolQ | Performance | × | × | × | × |
| CIFAR-100 baseline improvement | Performance | × | × | × | × |
| **Success Rate** | | **47% (14/30)** | **50% (15/30)** | **17% (5/30)** | **17% (5/30)** |

Table 3: Performance Comparison Across AI Agents

| Task | Token Cost ($) | | Total Tokens | |
|---|---|---|---|---|
| | **ReAct** | **OH** | **ReAct** | **OH** |
| Vision finetuning - classification | 0.176 | 0.124 | 67,034 | 54,434 |
| ChannelViT | 1.06 | 0.215 | 338,055 | 177,182 |
| ChannelViT - Easy | 1.090 | 0.318 | 352,141 | 267,508 |
| ChannelViT - No tests | 0.091 | 0.121 | 33,208 | 53,475 |
| Dataset download - dataset does not exist | 0.018 | 0.051 | 13,735 | 16,125 |
| Dataset preprocessing | 0.078 | 0.103 | 27,210 | 42,277 |
| Model forward pass - shape mismatches | 0.069 | 0.075 | 27,629 | 44,396 |
| Pretrained model download - HuggingFace | 0.096 | 0.063 | 36,592 | 24,738 |
| CIFAR-10 long tailed | 0.089 | 0.334 | 29,659 | 351,943 |
| Fix problems in model and dataloader | 0.376 | 0.556 | 146,826 | 785,182 |
| NaN losses | 0.124 | 0.212 | 40,385 | 193,801 |
| Dataset download - Noisy Imagenette | 0.129 | 0.068 | 67,426 | 25,429 |
| Correct norm for pretrained model | 0.380 | 0.265 | 136,831 | 313,149 |
| Overfit on small dataset | 0.093 | 0.133 | 25,642 | 52,791 |
| Large training logs | 0.023 | 0.185 | 35,189 | 114,903 |
| Segmentation | 0.118 | 0.383 | 39,662 | 395,127 |
| Pretrained model download - Torchvision | 0.058 | 0.044 | 22,289 | 30,596 |
| CIFAR10 Training | 0.209 | 0.288 | 71,409 | 253,445 |
| Model Training - shape mismatches | 0.289 | 0.147 | 115,568 | 92,262 |
| Add implementation - VAR | 0.051 | 0.139 | 12,473 | 59,863 |
| Wandb integration | 0.155 | 0.258 | 65,810 | 266,738 |
| TinyBERT Eval | 0.313 | 0.573 | 121,773 | 762,780 |
| ViT debugging | 0.277 | 1.496 | 84706 | 2,526,426 |
| BoolQ | 0.343 | 0.650 | 131,458 | 993,175 |
| Improve CIFAR-10 baseline - existing model ckpt | 0.115 | 0.401 | 29,232 | 366,660 |
| Noisy Imagenette | 0.582 | 0.288 | 192,275 | 253,445 |
| Multi-head Latent Attention | 3.164 | 1.910 | 1,022,125 | 1,706,146 |
| Multi-head Latent Attention - no hidden tests | 0.160 | 1.278 | 40,068 | 380,506 |
| Proximal Policy Optimization | 0.369 | 3.048 | 87,988 | 904,267 |
| CIFAR-100 baseline improvement | 0.123 | 3.531 | 30,493 | 1,133,630 |

Table 4: Comparison of Token Metrics between ReAct-Sonnet and OpenHands-Sonnet

