# OpenReview forum: "ML-Dev-Bench: Comparative Analysis of AI Agents on ML development workflows"
_ICLR.cc/2025/Workshop/AgenticAI — ICLR 2025 Workshop AgenticAI Reject_

### Official Review · Reviewer_vKoR · 2025-03-03

**Rating:** 3
**Confidence:** 4

**Review:**

The paper presents ML-Dev-Bench, a benchmark designed to evaluate AI agents on full ML development workflows, covering tasks such as dataset handling, model training, debugging, API integration, model implementation, and performance optimization. The paper shows experimental results on 30 tasks to evaluate three agents ReAct, OpenHands, and AIDE.

My concerns about this paper are as follows:

1.	The introduction is relatively brief and does not sufficiently introduce the research problem. It would benefit from a clearer explanation of the challenges in ML development that AI agents face, as well as a more structured discussion of existing research gaps to help readers better understand the motivation behind this benchmark.

2.	In lines 34-36 of the introduction, the authors mention that previous work failed to 'integrate with third-party tools, debug complex issues that span multiple components of the ML pipeline, and balance trade-offs like model performance and cost to come up with optimal design.' However, it is unclear how the proposed benchmark directly addresses these specific challenges.

3.	As a benchmark paper, the study evaluates only three agents (ReAct, OpenHands, and AIDE). Incorporating additional agents, such as ResearchAgent [1], and testing against a broader range of LLMs, including models like Gemini, would provide a more comprehensive evaluation of current AI capabilities.
[1] Qian Huang, Jian Vora, Percy Liang, and Jure Leskovec. MLAgentBench: Evaluating Language Agents on Machine Learning Experimentation. In Forty-first International Conference on Machine Learning, June 2024.

4.	The benchmark relies on a binary success/failure metric, which may not capture the task difficulty and partial success. According to Table 3, all agents (ReAct-Sonnet, OHSonnet, Aide-4o, and ReAct-4o) fail on 15 out of 30 tasks, and none succeed in performance optimization tasks, making it difficult to draw meaningful conclusions about their comparative effectiveness. I suggest a fine-grained metric to evaluate them.

5.	Since LLM-based agents exhibit stochastic behavior, their performance can vary across multiple runs. However, the paper does not report results on repeated runs.

---

### Official Review · Reviewer_NZE1 · 2025-03-03
**Review for ML-Dev-Bench**

**Rating:** 3
**Confidence:** 4

**Review:**

ML-DEV-BENCH benchmarked 3 agents (ReAct, Openhands, and AIDE) on ML development workflows consisting of 30 tasks. These tasks including Dataset Handing, Model Training, Debugging, Model Implemetation, API Integration and Performance. The results indicates that Openhands with Claud Sonnet perform best.



Strengths:

The included tasks are comprehensive, which contains 30 tasks with 6 categories.

The results can clearly show the results.



Weaknesses:

* Insight: the insight of the design of the 30 tasks is not clear.  The questions like why they contain these tasks, what specific challenges they tend to address, why these categories are important and included, are not clearly presented.
* Results: The results only shows which method performs better, without deep analysis on it.
* The authors only contains 3 Comparison Agents, which is not enough to obtain a robust results.
*  The results should contain more figures but not only tables, which can increase the readability.

---

### Official Review · Reviewer_icoR · 2025-03-04

**Rating:** 6
**Confidence:** 3

**Review:**

This paper introduces ML-Dev-Bench, a novel benchmark designed to evaluate AI agents on the entire spectrum of machine learning development workflows, which encompasses real-world tasks such as dataset handling, model training, debugging, model implementation, API integration, and performance optimization. The benchmark comprises 30 carefully designed tasks and evaluates three different agents, providing a detailed analysis of their strengths and limitations in addressing practical ML development challenges. Additionally, the paper presents an evaluation framework called Calipers, which standardizes the testing process and quantifies performance through binary success metrics, token cost analyses, and comprehensive task breakdowns.

The paper presents a well-motivated and clearly structured benchmark that fills an important gap in current evaluations of AI agents for ML development. Its clarity is shown by detailed task descriptions and systematic categorization. In terms of originality, ML-Dev-Bench offers a fresh perspective by moving beyond isolated coding tasks to assess the end-to-end challenges inherent in real-world ML workflows, which is a contribution that is both novel and timely.

### Pros:

- This paper introduces a comprehensive evaluation framework tailored to real-world ML development workflows.

- This paper evaluates multiple AI agents, offering insights into their performance across diverse task types.

- Includes token cost analysis and granular success rates, which add depth to the evaluation.

### Cons:

- Additional insights into common failure modes and error patterns would strengthen the overall evaluation.

- Could benefit from a more in-depth discussion of potential limitations and directions for future work.

---

### Decision · Program_Chairs · 2025-03-05

Reject